# Quantitative 3D imaging of the cranial microvascular environment at single-cell resolution

Alexandra N. Rindone [1,2], Xiaonan Liu [3,4], Stephanie Farhat [5,6,7], Alexander Perdomo-Pantoja [8], Timothy F. Witham [8], Daniel L. Coutu [5,6,7], Mei Wan [3] & Warren L. Grayson [1,2,9,10,11✉]

Vascularization is critical for skull development, maintenance, and healing. Yet, there remains a significant knowledge gap in the relationship of blood vessels to cranial skeletal progenitors during these processes. Here, we introduce a quantitative 3D imaging platform to enable the visualization and analysis of high-resolution data sets (>100 GB) throughout the entire murine calvarium. Using this technique, we provide single-cell resolution 3D maps of vessel phenotypes and skeletal progenitors in the frontoparietal cranial bones. Through these high-resolution data sets, we demonstrate that CD31$^{hi}$Emcn$^{hi}$ vessels are spatially correlated with both Osterix+ and Gli1+ skeletal progenitors during postnatal growth, healing, and stimulated remodeling, and are concentrated at transcortical canals and osteogenic fronts. Interestingly, we find that this relationship is weakened in mice with a conditional knockout of PDGF-BB in TRAP+ osteoclasts, suggesting a potential role for osteoclasts in maintaining the native cranial microvascular environment. Our findings provide a foundational framework for understanding how blood vessels and skeletal progenitors spatially interact in cranial bone, and will enable more targeted studies into the mechanisms of skull disease pathologies and treatments. Additionally, our technique can be readily adapted to study numerous cell types and investigate other elusive phenomena in cranial bone biology.

[1] Department of Biomedical Engineering, Johns Hopkins University School of Medicine, Baltimore, MD, USA. [2] Translational Tissue Engineering Center, Johns Hopkins University School of Medicine, Baltimore, MD, USA. [3] Department of Orthopaedic Surgery, Johns Hopkins University School of Medicine, Baltimore, MD, USA. [4] Division of Orthopaedics and Traumatology, Department of Orthopaedics, Nanfang Hospital, Southern Medical University, Guangzhou, Guangdong, China. [5] Regenerative Medicine Program, Ottawa Hospital Research Institute, Ottawa, ON, Canada. [6] Department of Cellular and Molecular Medicine, University of Ottawa, Ottawa, ON, Canada. [7] Division of Orthopaedic Surgery, The Ottawa Hospital, Ottawa, ON, Canada. [8] Department of Neurosurgery, Johns Hopkins University School of Medicine, Baltimore, MD, USA. [9] Department of Materials Science and Engineering, Johns Hopkins University, Baltimore, MD, USA. [10] Department of Chemical and Biomolecular Engineering, Johns Hopkins University, Baltimore, MD, USA. [11] Institute for NanoBioTechnology, Johns Hopkins University, Baltimore, MD, USA. ✉email: wgrayson@jhmi.edu

Vascularization is essential for the development, growth, and maintenance of craniofacial bone. Blood vessels provide oxygen and essential nutrients, support hematopoiesis, and transport hormones to and from the bone[1]. During development, vessels provide a template for mineralization during intramembranous and endochondral ossification of craniofacial bones[2], and vascular abnormalities have been linked to craniofacial syndromes such as mandibular hypoplasia[3], hemifacial microsomia[4], cleft palate[5], and craniosynostosis[6]. Furthermore, angiogenesis is necessary for bone formation during distraction osteogenesis[7] and defect healing[8] in the skull. However, there remains a significant knowledge gap in the relationship between blood vessels and osteoprogenitors during craniofacial bone growth, remodeling, and healing.

The distribution of craniofacial vessel subtypes and their spatial relationship to skeletal progenitors remain poorly understood due to a lack of adequate imaging technologies. Methods used in long bone involve slicing the tissue into thick sections (~100 µm) along its long axis, enabling visualization of cellular structures throughout the different anatomical regions of the bone. However, these techniques are difficult to adapt to craniofacial bone due to its irregular, curved geometry. As an alternative to tissue sectioning, intravital microscopy has been used to evaluate the microvascular environment in the calvarium. Using this method, studies have demonstrated that the majority of osteoblasts reside in close proximity (<10 µm) to vessels within marrow cavities[9] and that vessels deep within the marrow space exhibit less blood flow and oxygen tension than those near the endosteum[10,11], an active site of bone remodeling. However, due to the inherent limitations of intravital microscopy, these studies were confined to imaging small regions (mm² areas) and their results could not be extrapolated to different regions of the calvarium.

Recent developments in optical tissue clearing and light-sheet microscopy are a promising avenue for characterizing the microvascular environment of large, irregularly shaped tissues such as the skull. Using these techniques, one can image bones in 3D at single-cell resolution[12,13], providing a means of assessing cellular microenvironments within different anatomical regions of bone. However, the specific methods employed by these studies are limited to using endogenous fluorescence for labeling cellular structures and would not be capable of visualizing different vessel and cell types simultaneously. Furthermore, there remains a lack of quantitative methods to assess the 3D spatial relationships of cellular structures in large light-sheet imaging data sets (>100 GB). These shortcomings impair the otherwise powerful ability of light-sheet imaging to study the relationship between vessel phenotypes and skeletal progenitors in the skull.

In this study, we have developed a 3D imaging platform to comprehensively study the microvascular environment in cranial bone. Our method enables quantitative characterization of blood vessels and bone cells in the murine calvarium by combining whole-mount immunostaining, optical tissue clearing, light-sheet microscopy, and advanced 3D image analysis. Using this platform, we provide single-cell resolution 3D maps of vessel subtypes and skeletal progenitors in the frontoparietal bones of the calvarium. Then, we study how the spatial distribution of vessel phenotypes and skeletal progenitors in the calvarium vary during postnatal bone growth, alterations in bone remodeling, and healing. Our findings provide a foundational framework that will progress our knowledge of craniofacial bone biology and inform the development of therapies to treat skull abnormalities and injuries.

## Results

### Quantitative 3D light-sheet imaging of murine calvaria. To visualize vessel phenotypes and skeletal progenitors in the murine

calvarium, we developed and optimized an imaging pipeline comprised of whole-mount immunostaining, optical tissue clearing, and light-sheet imaging (Supplementary Fig. 1a). We adapted a staining regimen used in long-bone sections[14] to achieve adequate antibody penetration and labeling of up to three molecular markers: CD31 and Emcn for vessels and Osterix or Gli1 for skeletal progenitor subpopulations. Osterix is a marker for skeletal progenitors that are restricted to the osteoblast lineage[15], while Gli1 marks less-differentiated skeletal stem cells[16,17]. To achieve high-quality visualization of each marker, we used fluorophores spanning the red to the infrared spectrum, as ultraviolet and green dyes resulted in high levels of background and light scattering during imaging. We cleared the calvaria by removing the blood prior to staining and incubating in a graded series of 2,2-thiodiethanol (TDE) following staining. This method did not require us to decalcify bone to achieve adequate bone tissue clearing and maintained the geometry of the calvarium (Supplementary Fig. 1B, C). Following clearing, we imaged calvaria using light-sheet microscopy, which allowed us to rapidly acquire high-resolution data and minimize photo-bleaching through the duration of the scan (~5 h per sample). Our resulting images captured the 3D distribution of different vessel phenotypes and skeletal progenitors in the calvarium with high axial and lateral resolution (Fig. 1A–D and Supplementary Video 1).

To analyze vessel phenotypes and skeletal progenitors, we developed a quantitative pipeline to enable 3D spatial characterization of high-resolution datasets (~500 GB raw, ~200 GB compressed, Supplementary Fig. 2A). We applied the spots and surfaces modules in Imaris software to segment blood vessels and skeletal progenitors, and then performed a series of masking and filtering algorithms to denote three-vessel phenotypes: CD31$^{hi}$Emcn$^{-}$ arteries and arterioles, CD31$^{hi}$Emcn$^{hi}$ capillaries, and CD31$^{lo}$Emcn$^{hi}$ capillaries and sinusoids[18,19] (Supplementary Fig. 2B–D). CD31$^{lo}$Emcn$^{lo}$ sinusoids were not segmented due to their low signal-to-noise ratio. We exported the spots and surfaces statistics into GraphPad Prism and XiT software[14] to analyze vessel volume, skeletal progenitor number, and vessel-skeletal progenitor spatial distances (Supplementary Fig. 2E). The resulting analysis provides a comprehensive characterization of the spatial coupling between vessels and skeletal progenitors by reflecting the native 3D environment across large tissue volumes (cm³).

**3D map of calvarial vessels and skeletal progenitors**. Using our imaging pipeline, we generated high-resolution 3D maps of vessel phenotypes, Osterix+ skeletal progenitors, and Gli1+ skeletal progenitors in the parietal and posterior frontal bones of juvenile 4-weeks-old mice. Vessels were located in the periosteum, dura mater, transcortical canals, marrow cavities, and osteogenic fronts adjacent to the sutures (Fig. 2A–D and Supplementary Fig. 3A, B). Marrow vessels in the parietal and frontal bones were observed near the sutures, while only periosteal and meningeal vessels were observed at the center of each bone. Vessel phenotypes were also differentially distributed in the calvarium: Most CD31$^{hi}$Emcn$^{-}$ arteries and arterioles were present in the periosteum and dura mater, while CD31$^{lo}$Emcn$^{hi/lo}$ sinusoids were restricted to the marrow cavities. CD31$^{hi}$Emcn$^{hi}$ capillaries were present in the periosteum, dura mater, marrow cavities, and osteogenic fronts, connecting CD31$^{hi}$Emcn$^{-}$ periosteal arterioles to CD31$^{lo}$Emcn$^{hi}$ sinusoids. Expression of CD31 and Emcn in CD31$^{hi}$Emcn$^{hi}$ capillaries was most intense at the transcortical canals—the regions that enable arterioles in the periosteum and dura mater to connect to venous sinusoids.

Similar to vessel phenotypes, skeletal progenitors varied in their spatial distribution. Osterix+ osteoprogenitors were prevalent in the periosteum and dura mater, osteogenic fronts nearby sutures, transcortical canals, and marrow cavities of the parietal

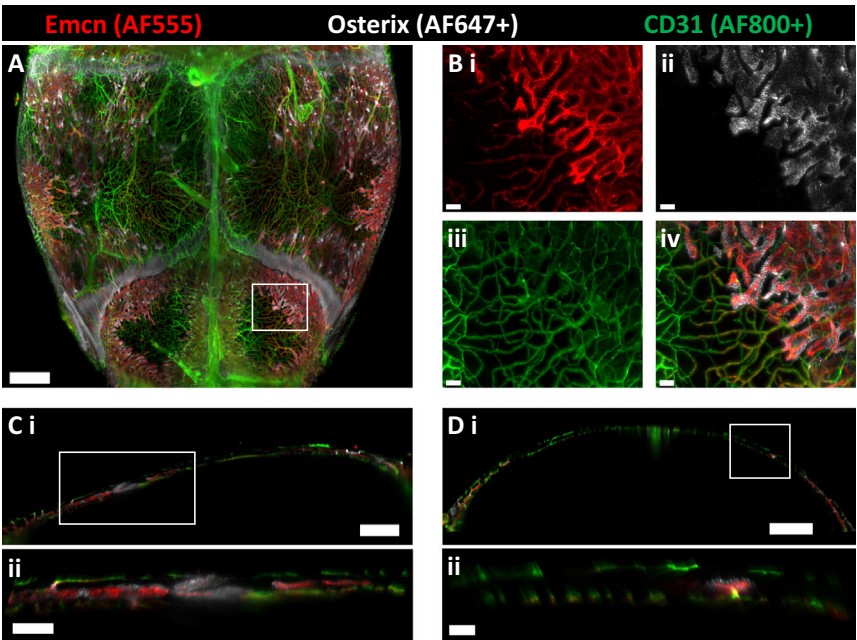

**Fig. 1 3D light-sheet imaging of the murine calvarium at single-cell resolution. A** Maximum intensity projection (MIP) of vessels and Osterix+ skeletal progenitors in the parietal and posterior frontal bones of the calvarium. **B** Images from the boxed region in **A** demonstrating the high resolution and signal quality obtained in each channel (i–iv). The dyes used for each channel are designated above (**A**) and (**B**): AF555 for Endomucin (i), AF647 plus for Osterix (ii), and AF800 plus for CD31 (iii). **C, D** Sagittal (**C**) and frontal (**D**) 40-μm-thick optical sections demonstrating the location of vessels and Osterix+ cells along the thickness of the calvarium. Boxed regions from (**C**i) and (**D**i) are shown in **C**ii and **D**ii. Similar results were achieved for all the calvaria imaged in this study ($n = 45$ calvaria). Scale bars: 1000 μm (**A, D**i), 800 μm (**C**i), 300 μm (**C**ii), 100 μm (**B**i–iv, **D**ii). Colors: Red: Endomucin (Emcn), Gray: Osterix, Green: CD31.

and frontal bones (Fig. 2A, C). Gli1+ progenitors—a marker for less-differentiated skeletal stem cells[16,17]—were concentrated at the sutures, transcortical canals, and marrow cavities adjacent to transcortical canals, but they were mostly absent from the periosteum and dura mater (Fig. 2B, D). Interestingly, expression of Gli1 was visibly more intense at the transcortical canals compared to the sutures.

To determine whether skeletal progenitors exhibited a preferential spatial relationship to specific vessel phenotypes, we quantified the distribution of Osterix+ and Gli1+ progenitors relative to each vessel type. We found that both progenitor populations were preferentially associated with CD31$^{hi}$Emcn$^{hi}$ vessels compared to other vessel phenotypes (Fig. 2E). This relationship was most apparent at the transcortical canals, where we observed the highest protein expression of CD31 and Emcn in vessels and Osterix or Gli1 in skeletal progenitors.

**Postnatal growth shifts the distribution of vessel phenotypes and Osterix+ progenitors.** Next, we compared the calvaria of juvenile (4-weeks-old) and adult (12-weeks-old) mice to determine how vessel phenotype and skeletal progenitor distribution change following postnatal growth. In adult mice, there were fewer CD31$^{hi}$Emcn$^{-}$ periosteal and meningeal vessels and visible increases in CD31$^{lo}$Emcn$^{lo}$ sinusoids (not quantified due to low fluorescence intensity, Fig. 3A–F, I). These changes were corroborated by microCT data, which showed greater development of bone marrow cavities in adult calvaria (Fig. 3K–M). There were no significant changes in the volume of CD31$^{hi}$Emcn$^{hi}$ or CD31$^{lo}$Emcn$^{hi}$ sinusoids, although fewer CD31$^{hi}$Emcn$^{hi}$ vessels were observed in the periosteum and dura mater (Fig. 3A–F, I). Additionally, the total vessel volume remained the same (Fig. 3H).

Along with changes in vessel phenotypes, the numbers of Osterix+ and Gli1+ skeletal progenitors decreased in adult

calvaria, and their distribution was mainly restricted to the sutures, transcortical canals, and bone marrow cavities (Fig. 3A–F, G). While a decrease in progenitors was observed in different regions of the calvarium, the most significant decline occurred in the parietal bones (Supplementary Fig. 4A). More-over, Osterix+ cells were mostly absent in the periosteum and dura mater of adult calvaria (Fig. 3A–C). These results correlated with differences in vessel-progenitor relationships: The fraction of Osterix+ cells within 5 μm of the nearest vessel in adult versus juvenile calvaria was significantly higher for CD31$^{hi}$Emcn$^{hi}$ and CD31$^{lo}$Emcn$^{hi}$ vessels and lower for CD31$^{hi}$Emcn$^{-}$ vessels (Fig. 3J). These trends held across different regions of the calvarium (Supplementary Fig. 4B). There were no significant changes in the relationship of Gli1+ cells to vessel phenotypes between juvenile and adult calvaria (Fig. 3J). Nonetheless, both progenitor cell types maintained a preferential spatial association with CD31$^{hi}$Emcn$^{hi}$ vessels at 4 and 12 weeks of age.

**PTH stimulates Osterix+ progenitor proliferation, but does not alter vessel phenotype distribution.** To provide insight on how vessel phenotypes and skeletal progenitors interact during calvarial bone remodeling, we administered a parathyroid hormone analog (PTH 1–34) daily for 1 month, a regimen previously shown to increase osteoblast number and bone mineral deposition in murine long bone[20]. We found that PTH administration did not significantly change the fractional volume for each vessel phenotype or total vessel volume; although, there were areas of increased Emcn signal intensity in sinusoids near the transcortical canals (Fig. 4A–D, G–J, K, L). Furthermore, we observed increased marrow cavities and CD31$^{lo}$Emcn$^{lo}$ sinusoids in the parietal bone with PTH administration—a finding complemented by decreased bone volume to total volume percentage (BV/TV) and increased bone surface area (SA) (Fig. 4O–Q).

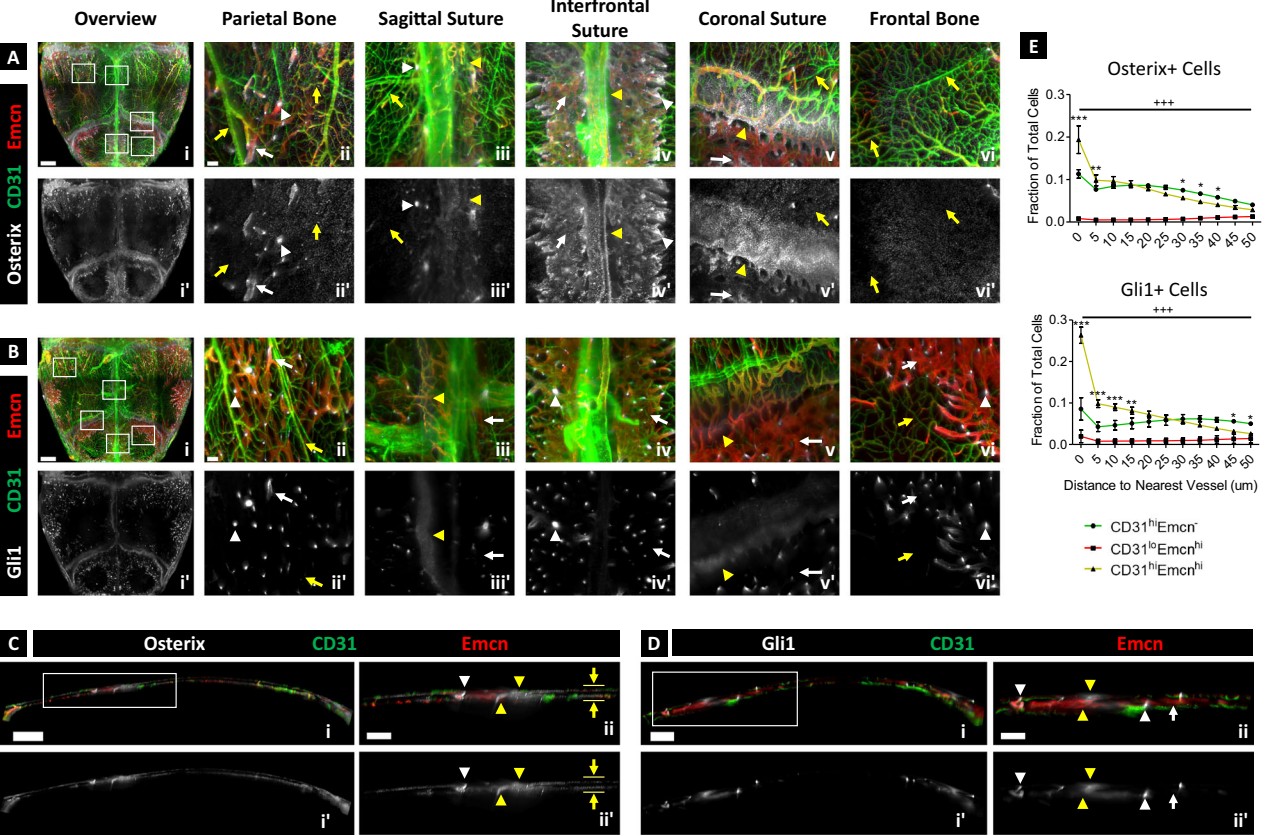

**Fig. 2 3D map of the microvascular environment in 4-weeks-old murine calvaria. A**, **B** MIP displaying calvarial vessels (**A**i, **B**i) and Osterix+ (**A**i, **A**i') or Gli1+ skeletal progenitors (**B**i, **B**i') in the parietal and posterior frontal bones of the calvarium. Zoomed-in images demonstrate the distribution of vessel phenotypes (**A**ii–vi, **B**ii–vi) and skeletal progenitors (**A**ii–vi, **A**ii'–vi', **B**ii–vi, **B**ii'–vi') in different regions. **C**, **D** 50-μm-thick sagittal sections showing the locations of vessels **C**i–ii, **D**i–ii) and skeletal progenitors (**C**i–ii, **C**i'–ii', **D**i–ii, **D**i'–ii') along the thickness of the calvarium. CD31$^{hi}$Emcn$^-$ and CD31$^{hi}$Emcn$^{hi}$ vessels are prevalent in the periosteum and dura mater (yellow arrows), while CD31$^{hi}$Emcn$^{hi}$ and CD31$^{lo}$Emcn$^{hi}$ vessels make up the majority of vessels in the bone marrow cavities (white arrows). Osterix+ progenitors (**A**, **C**) are distributed throughout the periosteum and dura mater (yellow arrows), transcortical canals (white arrowheads), and osteogenic fronts nearby the sutures (yellow arrowheads), while Gli1+ progenitors (**B**, **D**) are mainly restricted to the transcortical canals and sutures. Results were replicated in 3 calvaria for each staining combination (CD31/Emcn/Osterix, CD31/Emcn/Gli1). **E** Plots showing the nearest distance of Osterix+ (top) and Gli1+ (bottom) progenitors to each vessel phenotype (n = 3). Both cell types exhibit a preferential spatial relationship to CD31$^{hi}$Emcn$^{hi}$ vessels compared to the other phenotypes. CD31$^{lo}$Emcn$^{lo}$ vessels were not quantified due to their low fluorescence signal-to-noise ratio. Data are mean ± SD. Statistics were performed using a two-way ANOVA and Bonferroni post-hoc test. ***$p < 0.001$, **$p < 0.01$, *$p < 0.05$ between CD31$^{hi}$Emcn$^{hi}$ and CD31$^{hi}$Emcn$^-$ vessels; +++$p < 0.001$ between CD31$^{lo}$Emcn$^{hi}$ and CD31$^{hi}$Emcn$^-$ vessels; Scale bars: 1000 μm (**A**i, **B**i), 100 μm (**A**ii–vi, **B**ii–vi), 700 μm (**C**i), 300 μm (**C**ii), 500 μm (**D**i), 200 μm (**D**ii). Colors: Red: Endomucin (Emcn), Gray: Osterix (**A**, **C**) or Gli1 (**B**, **D**), Green: CD31.

Despite a lack of significant change in vessel phenotypes, we found differences in the skeletal progenitor populations with PTH administration. PTH significantly increased the total number of Osterix+ progenitors, especially in the periosteum and dura mater (Fig. 4A–F, M). By contrast, PTH did not increase the number of Gli1+ progenitors (Fig. 4G–J, M). However, there were some changes in Gli1+ progenitor distribution. Cells moderately expressing Gli1 expanded in the marrow cavities adjacent to transcortical canals—particularly near vessels with high Emcn expression—and in periosteal and transcortical canals nearby the coronal suture (Fig. 4J). Both Osterix+ and Gli1+ progenitors remained preferentially associated with CD31$^{hi}$Emcn$^{hi}$ vessels following PTH administration, but the fraction of Osterix+ cells within 5 μm of a CD31$^{hi}$Emcn$^{hi}$ vessel was significantly reduced compared to the control (Fig. 4N).

**Loss of preosteoclast PDGF-BB secretion decreases the spatial affinity of skeletal progenitors to CD31$^{hi}$Emcn$^{hi}$ vessels.** Preosteoclast-derived PDGF-BB is required for angiogenesis with coupled osteogenesis during normal bone homeostasis and in disease

conditions[21,22]. To determine the phenotypic changes in calvarial blood vessels and skeletal progenitors in response to decreased bone remodeling activity, we used Trap+ osteoclast lineage-specific conditional *Pdgfb* deletion mice (Pdgfb$^{cKO}$) by crossing *Trap-Cre* mice with *Pdgfb*-floxed mice. In the calvaria of 4-weeks-old mice, we found that CD31$^{hi}$Emcn$^{hi}$ fractional volume and total vessel volume decreased, while fractional CD31$^{hi}$Emcn$^-$ vessel volume increased in Pdgfb$^{cKO}$ mice relative to *Pdgfb*-floxed (WT) mice (Fig. 5A–H, K–L). While we did not find any statistical differences in the number of Osterix+ or Gli1+ cells, we found that the preferential association of these cells to CD31$^{hi}$Emcn$^{hi}$ vessels was significantly reduced in Pdgfb$^{cKO}$ calvaria (Fig. 5I–J, M). This effect was most apparent at the transcortical canals and osteogenic fronts, where the concentration of Osterix+ and Gli1+ cells was visibly lower in Pdgfb$^{cKO}$ versus WT calvaria (Fig. 5C, D, G–H). We also observed alterations in bone microarchitecture: There was less bone marrow cavity development in the frontal bones of Pdgfb$^{cKO}$ calvaria, as demonstrated by a higher BV/TV percentage and lower bone SA (Fig. 5N–P). To determine whether osteoclasts resided in close proximity to these regions, we stained for Vpp3—a marker known to exclusively stain osteoclasts in bone[23]. Most osteoclasts were found adjacent to

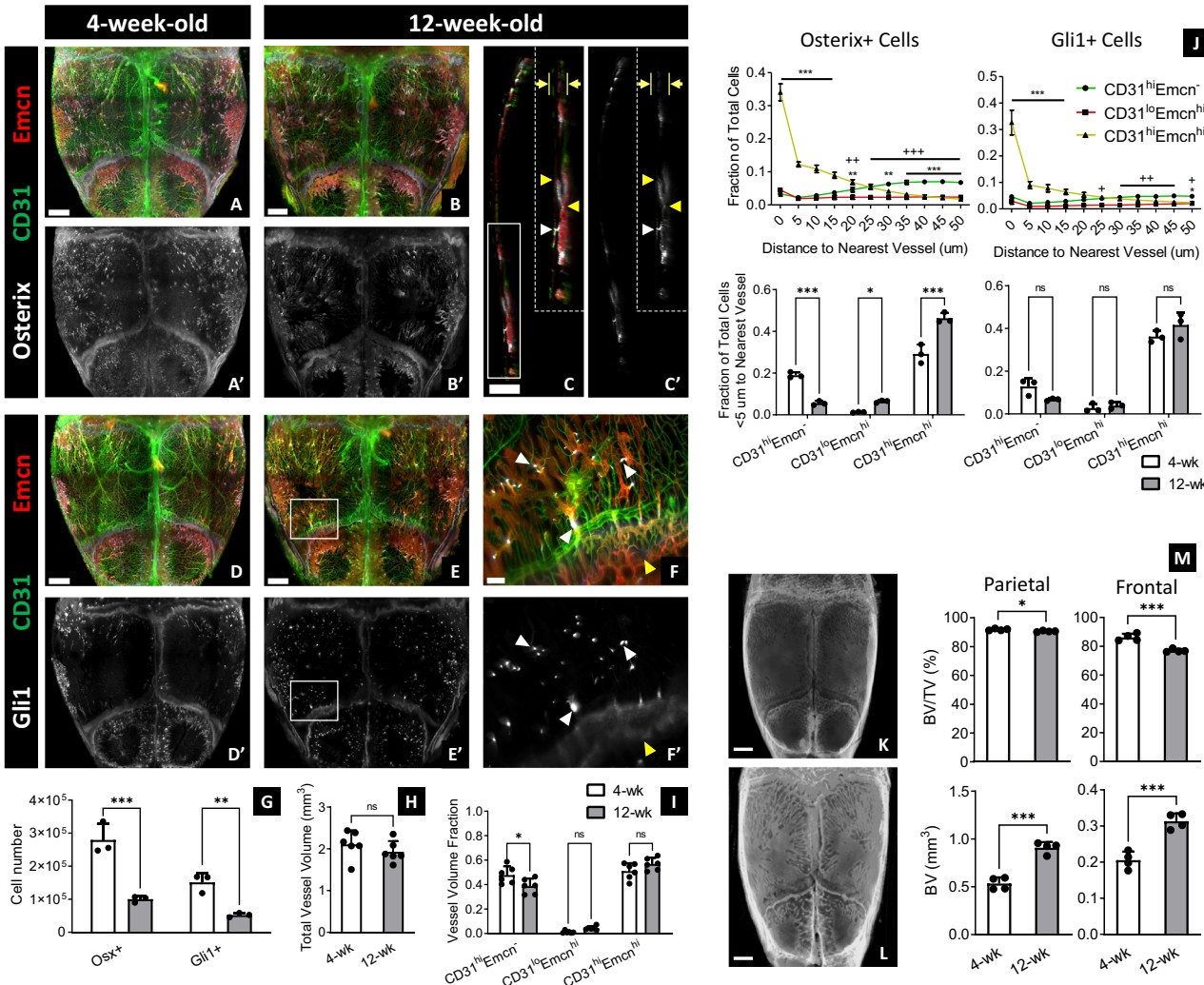

**Fig. 3 Changes in vessel phenotype and skeletal progenitor distribution following postnatal development. A**, **B** MIP of vessels and Osterix+ progenitors from calvaria of 4-weeks-old (**A**, **A'**) and 12-weeks-old (**B**, **B'**) mice. **C** 40-μm-thick sagittal section showing vessels (**C**) and Osterix+ progenitors (**C**, **C'**) along the thickness of the calvarium displayed in **B**. 12-weeks-old calvaria have more bone marrow vessels (CD31$^{hi}$Emcn$^{hi}$ and CD31$^{lo}$Emcn$^{hi/lo}$) and fewer periosteal and meningeal vessels compared to 4-weeks-old calvaria. Osterix+ progenitors in 12-weeks-old calvaria are spatially restricted to transcortical canals (white arrowheads), bone marrow cavities, and osteogenic fronts adjacent to the sutures (yellow arrowheads), and are absent from the periosteum and dura mater (yellow arrows). **D**, **E** MIP of vessels and Gli1+ progenitors of calvaria from 4-weeks-old (**D**) and 12-weeks-old (**E**) mice. **F** Zoomed-in region from **E**. Gli1+ cells in 12-weeks-old calvaria (**E**, **E'**, **F**, **F'**) are concentrated at and nearby the transcortical canals (white arrowheads) and the sutures (yellow arrowheads). Results were replicated in 3 calvaria for each experimental group and staining combination (12 total calvaria). **G–I** Comparison of total skeletal progenitor number (**G**), total vessel volume (**H**), and fractional vessel phenotype volumes (**I**) between 4- and 12-weeks-old calvaria (n = 3 for **G**; n = 6 for **H**, **I**). **J** Plots representing the spatial correlation of Osterix+ and Gli1+ progenitors to vessel phenotypes (n = 3). Both cell types maintain a preferential relationship with CD31$^{hi}$Emcn$^{hi}$ vessels in 12-weeks-old mice. Osterix+ cells demonstrate a higher spatial affinity to CD31$^{hi}$Emcn$^{hi}$ vessels in 12-weeks-old versus 4-weeks-old calvaria. **K**, **L** MicroCT 3D volume projections of calvaria at 4 weeks (**K**) and 12 weeks (**L**) of age. **M** MicroCT quantification of bone volume to total volume (BV/TV) percentage and bone volume (BV) in the parietal and posterior frontal bones of 4- and 12-weeks-old mice (n = 4). Data are mean ± SD. Statistics were performed using a two-way ANOVA with Bonferroni post-hoc test (**G–J**) or two-tailed t-test (**H**, **M**). ***$p < 0.001$, **$p < 0.01$, *$p < 0.05$ where designated or between CD31$^{hi}$Emcn$^{hi}$ and CD31$^{hi}$Emcn$^-$ vessels in **J**; +++$p < 0.001$ between CD31$^{lo}$Emcn$^{hi}$ and CD31$^{hi}$Emcn$^-$ vessels (**J**). Exact p-values for two-tailed t-tests: **H** $p = 0.3321$, **M** $p = 0.0213$ (top left), 0.0005 (top right), 0.0001 (bottom left), 0.0005 (bottom right). Scale bars: 1000 μm (**A**, **B**, **D**, **E**, **K**, **L**); 700 μm (**C**); 300 μm (**C**, inset); 200 μm (**F**). Colors: Red: Endomucin (Emcn), Gray: Osterix (**A–C**) or Gli1 (**D–F**), Green: CD31.

CD31$^{hi}$Emcn$^{hi}$ and CD31$^{lo}$Emcn$^{hi}$ vessels in the marrow cavities and within proximity to the transcortical canals (Supplementary Fig. 5A–C).

**CD31$^{hi}$Emcn$^{hi}$ vessels and Gli1+ progenitors infiltrate into calvarial defect following injury.** In addition to remodeling, we

investigated the contribution of vessel phenotypes and skeletal progenitors to calvarial bone healing. We created 1-mm subcritical-sized defects in the parietal bone of adult mice and evaluated healing at 21- and 56-days following fracture (PFD21, PFD56; PFD: post-fracture day). At PFD21, defects were highly vascularized, and the majority of vessels were CD31$^{hi}$Emcn$^{hi}$ (Fig. 6A, B, E, F, L, N). Gli1+ cells were highly concentrated across the entire defect region, while Osterix+

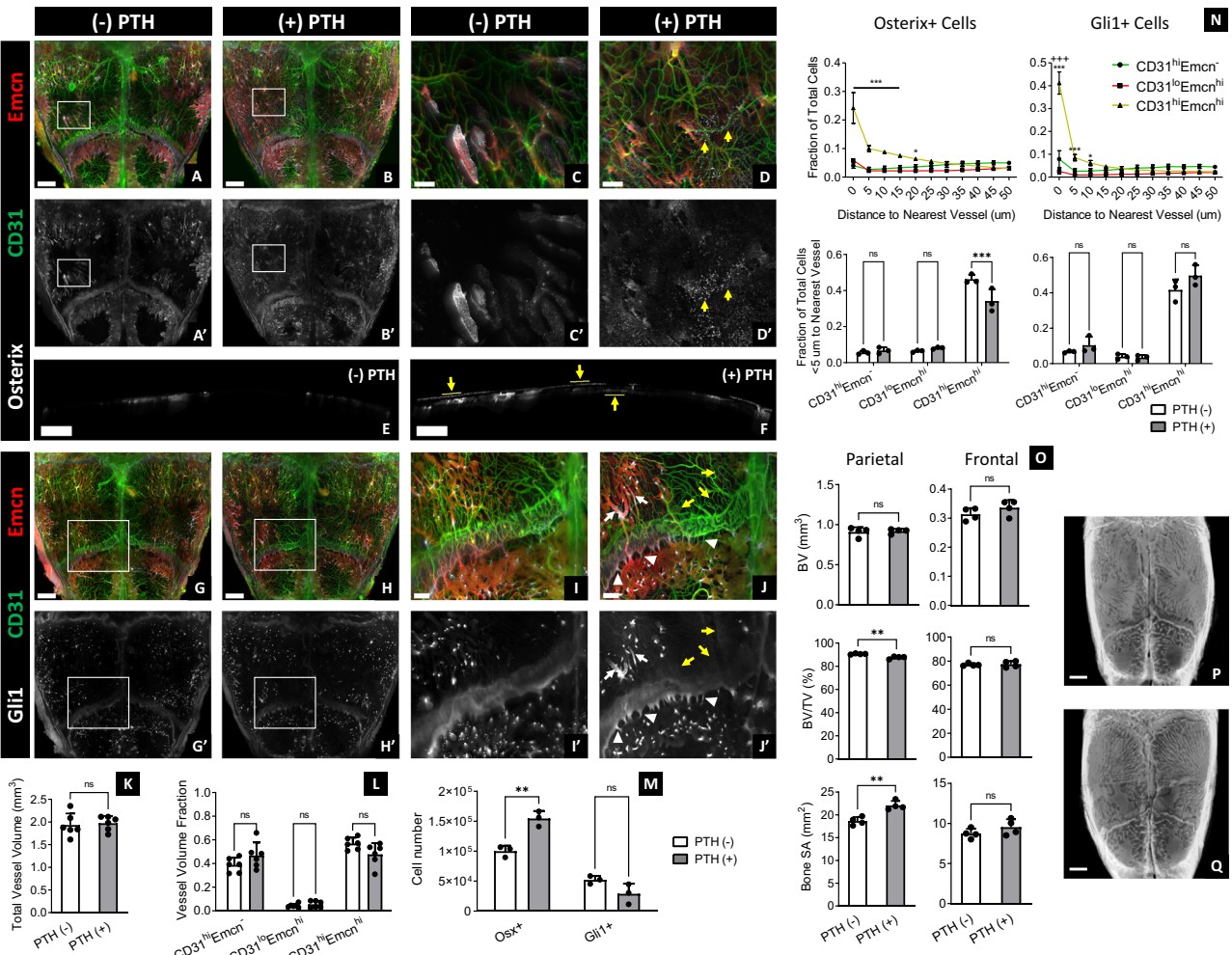

**Fig. 4 PTH administration increases the number of Osterix+ progenitors, but does not significantly affect Gli1+ progenitor number or CD31^hiEmcn^hi vessel volume. A–D** MIP of calvarial vessels (**A–D**) and Osterix+ progenitors (**A–D, A'–D'**) in 12-weeks-old mice with (**B, B', D, D'**) or without (**A, A', C, C'**) 4 weeks of PTH administration. Increased Osterix+ cells are present within the periosteum and dura mater of PTH-treated calvaria (yellow arrows). **E–J** 40-μm-thick sagittal sections demonstrating the expansion of Osterix+ progenitors in the periosteum and dura mater (yellow arrows) in PTH-treated calvaria (**E**) compared to the control (**F**). **G–J** MIP of calvarial vessels (**G–J**) and Gli1+ skeletal progenitors (**G–J, G'–J'**) in control (**G, G', I, I'**) and PTH-treated (**H, H', J, J'**) groups. More Gli1+ cells are observable in the marrow cavities nearby the transcortical canals (white arrows), as well as in the transcortical canals (white arrowheads) and periosteum (yellow arrows) nearby the sutures in PTH-treated calvaria. Results were replicated in 3 calvaria for each experimental group and staining combination (12 total calvaria). **K–M** Total vessel volume (**K**), fractional vessel phenotype volume (**L**), and skeletal progenitor number (**M**) in control and PTH-treated calvaria (*n* = 3 for **M**; *n* = 6 for **K, L**). **N** Plots representing the spatial correlation of Osterix+ and Gli1+ progenitors to vessel phenotypes (*n* = 3). Both cell types maintain a preferential relationship with CD31^hiEmcn^hi vessels following PTH treatment, but fewer Osterix+ cells are associated with CD31^hiEmcn^hi vessels compared to control calvaria. **O** MicroCT quantification of bone volume (BV), bone volume to tissue volume (BV/TV) percentage, and bone surface area (SA) in the parietal and posterior frontal bones of control and PTH-treated calvaria (*n* = 4). **P, Q** MicroCT 3D volume projections of control (**P**) and PTH-treated calvaria (**Q**). Data are mean ± SD. Statistics were performed using a two-way ANOVA with Bonferroni post-hoc test (**L–N**) or two-tailed *t*-test (**K, O**). ****p* < 0.001, ***p* < 0.01, **p* < 0.05 where designated or between CD31^hiEmcn^hi and CD31^hiEmcn^− vessels in **N**; +++*p* < 0.001 between CD31^loEmcn^hi and CD31^hiEmcn^− vessels (**N**). Exact *p*-values for two-tailed *t*-tests: **K** *p* = 0.7493, **O** *p* = 0.7771 (top left), 0.2299 (top right), 0.0021 (middle left), 0.7862 (middle right), 0.0021 (bottom left), 0.2111 (bottom right). Scale bars: 1000 μm (**A, B, G, H, P, Q**); 700 μm (**E, F**); 300 μm (**I, J**); 200 μm (**C, D**). Colors: Red: Endomucin (Emcn), Gray: Osterix (**A–F**) or Gli1 (**G–J**), Green: CD31.

cells resided in regions of active bone formation (Fig. 6A, B, E, F and Supplementary Fig. 6A). In addition, there was a substantial expansion of Osterix+ and Gli1+ cells in the periosteum extending from the defect to nearby sutures (Fig. 6A, B, E, F, I, J). This effect was unique to the periosteum, as there were few Osterix+ and Gli1+ cells detected in the dura mater—the only layer that remained uninjured following the creation of the defect (Fig. 6I, J). By PFD56, total vessel volume, fractional CD31^hiEmcn^hi volume, Osterix+ cell number, and Gli1+ cell number decreased in the defect relative to PFD21, but the vessel and Gli1+ cell density remained higher relative to the

surrounding uninjured bone (Fig. 6C, D, G, H, L–N). Furthermore, there was no significant change in defect bone volume, suggesting that most healing happened within the first 3 weeks of injury (Fig. 6O and Supplementary Fig. 6A, B). Nevertheless, both Osterix+ and Gli1+ progenitors remained preferentially associated with CD31^hiEmcn^hi vessels at PFD21 and PFD56 (Fig. 6K).

Since there was a significant expansion of skeletal progenitors around the defect region, we evaluated whether there was a systemic response to injury. We quantified skeletal progenitors in the ipsilateral and contralateral sides of the parietal bone

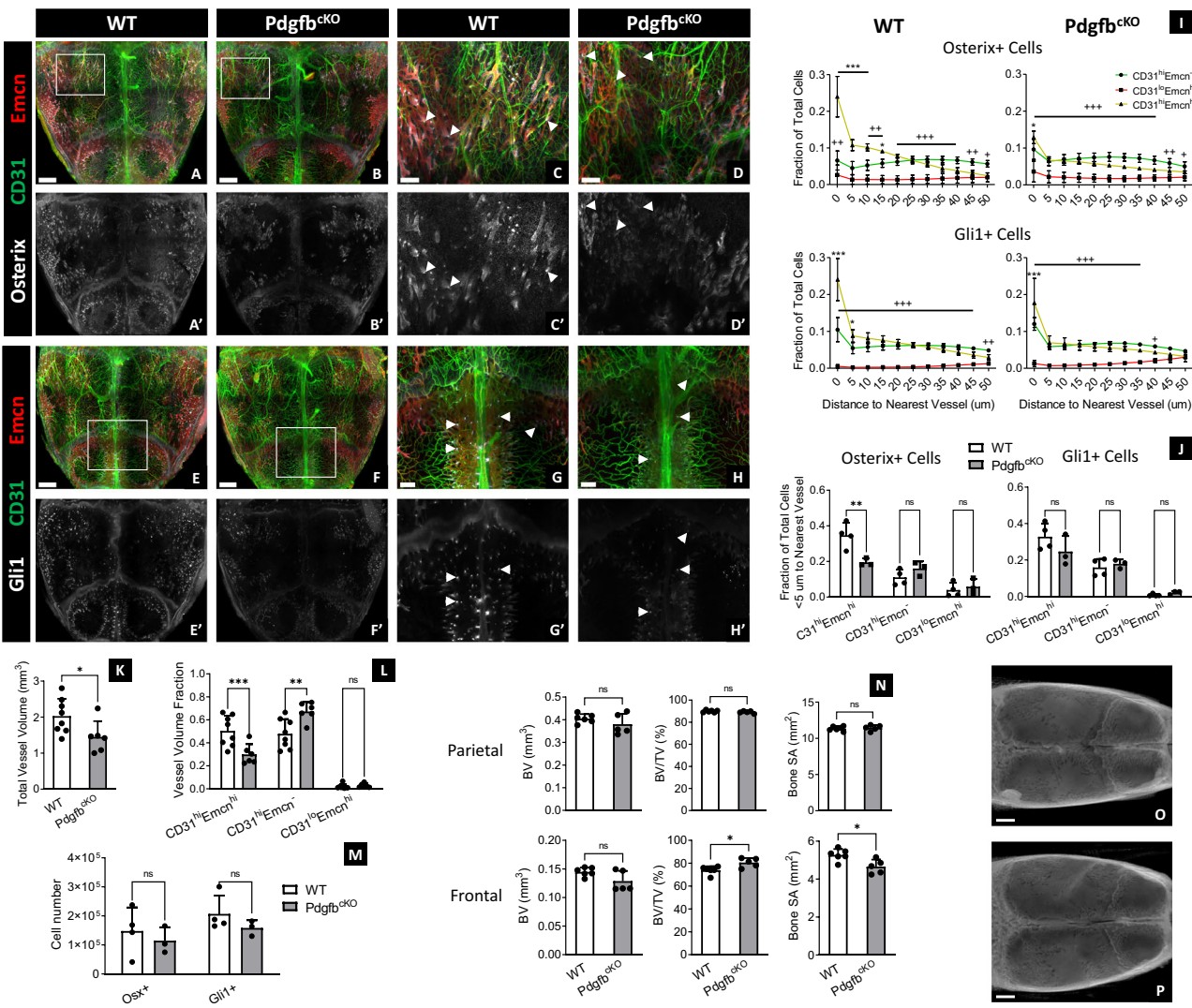

**Fig. 5 Knocking out PDGF-BB expression in TRAP+ osteoclasts decreases CD31^hiEmcn^hi vasculature and their association with Osterix+ and Gli1+ skeletal progenitors.** A–H MIP of vessels (A–H), Osterix+ progenitors (A–D, A'–D'), and Gli1+ progenitors (E–H, E'–H') from the calvaria of 4-weeks-old WT and Pdgfb^cKO mice. Protein expression of CD31 and Emcn in vessels and Osterix or Gli1 in skeletal progenitors is markedly reduced in Pdgfb^cKO calvaria at the transcortical canals (arrowheads) and osteogenic fronts nearby the sutures. Results were replicated in 4 WT calvaria and 3 Pdgfb^cKO calvaria for each staining combination (14 total calvaria). I Spatial distribution of Osterix+ and Gli1+ cells relative to each vessel phenotype in WT and Pdgfb^cKO calvaria (WT: $n = 4$ Osterix, $n = 4$ Gli1; Pdgfb^cKO: $n = 3$ Osterix, $n = 3$ Gli1). J Fraction of Osterix+ and Gli1+ progenitors within 5 μm of their nearest vessel in WT and Pdgfb^cKO calvaria (WT: $n = 4$ Osterix, $n = 4$ Gli1; Pdgfb^cKO: $n = 3$ Osterix, $n = 3$ Gli1). Significantly fewer Osterix+ cells are associated with CD31^hiEmcn^hi vessels in Pdgfb^cKO calvaria compared to the WT control. K–M Total vessel volume (K), fractional vessel phenotype volume (L), and skeletal progenitor number (M) in the calvaria of WT versus Pdgfb^cKO mice ($n = 8$ for WT and $n = 6$ for Pdgfb^cKO in K–L; M: $n$ is the same as in I and J). N MicroCT quantification of bone volume (BV), bone volume to tissue volume (BV/TV) percentage, and bone surface area (SA) in the parietal and posterior frontal bones of WT and Pdgfb^cKO calvaria (WT: $n = 6$; Pdgfb^cKO: $n = 5$). O, P MicroCT 3D volume projections of WT (O) and Pdgfb^cKO calvaria (P). Data are mean ± SD. Statistics were performed using a two-way ANOVA with Bonferroni post-hoc test (I, J, L, M) or two-tailed t-test (K, N). ***$p < 0.001$, **$p < 0.01$, *$p < 0.05$ where designated or between CD31^hiEmcn^hi and CD31^hiEmcn^− vessels in I; +++$p < 0.001$ between CD31^loEmcn^hi and CD31^hiEmcn^− vessels (I). Exact p-values for two-tailed t-tests: K $p = 0.0112$, N $p = 0.2565$ (top left), 0.0736 (bottom left), 0.2183 (top middle), 0.0152 (bottom middle), 0.6635 (top right), 0.0151 (bottom right). Scale bars: 1000 μm (A, B, E, F, O, P); 300 μm (C, D, G, H). Colors: Red: Endomucin (Emcn), Gray: Osterix (A–D) or Gli1 (E–H), Green: CD31.

and compared them to the number of progenitors in uninjured adult mice. Surprisingly, there were elevated levels of Gli1+ cells in both the ipsilateral and contralateral sides of the injured calvaria at PFD21 compared to the uninjured calvaria (Supplementary Fig. 6D–S). Most of this expansion occurred in the periosteum, especially in the regions near the sutures (Supplementary Fig. 5D, G, J, M). By PFD56, Gli1+ and Osterix+ cell number significantly decreased to levels comparable to the uninjured calvaria (Supplementary Fig. 6P–S).

However, alterations in bone surface topography and regions of excess mineral formation remained at PFD56 (Supplementary Fig. 6A–C).

## Discussion

Intimate spatial interactions between blood vessels and skeletal progenitors are essential for proper bone growth, remodeling, and

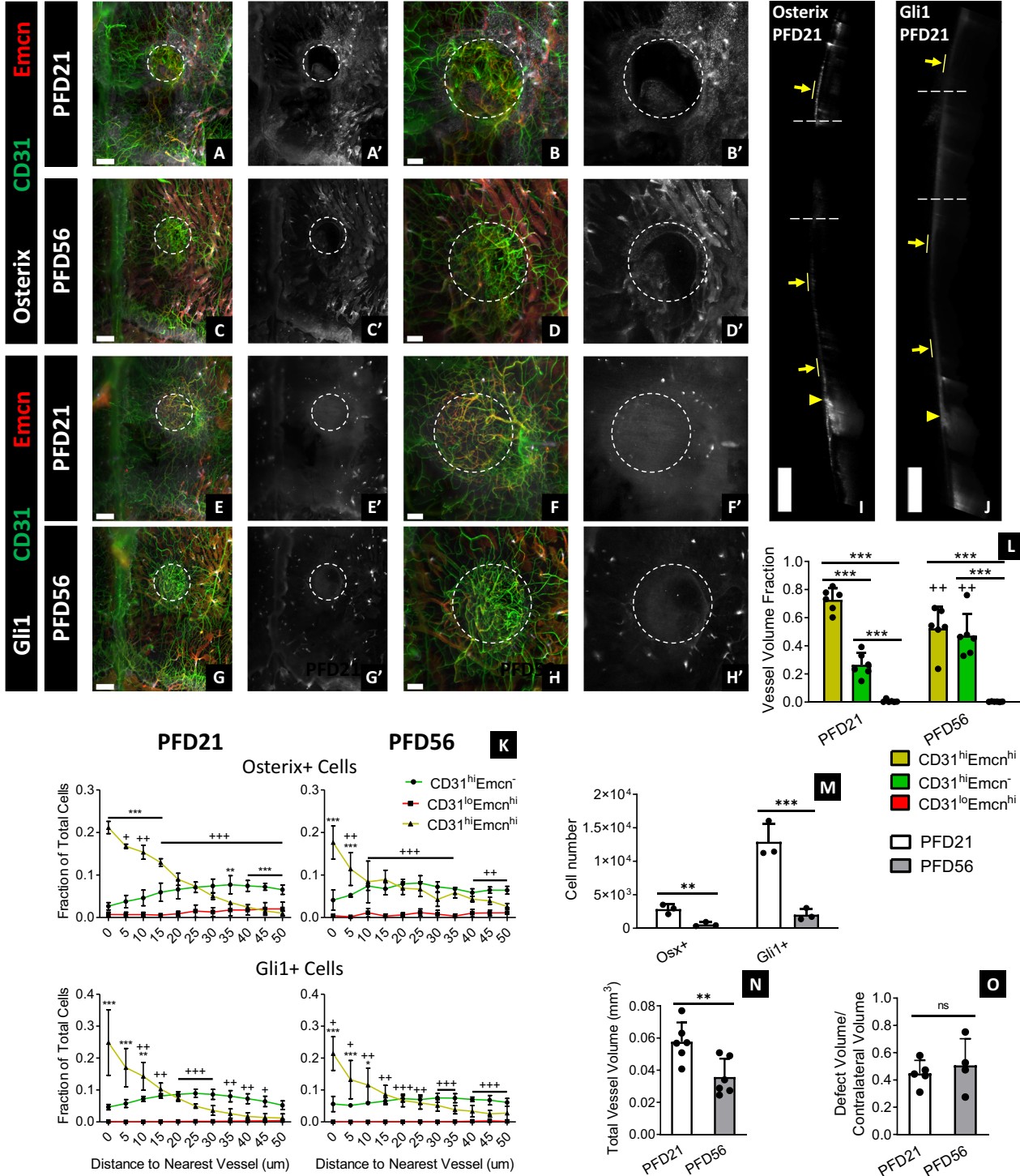

**Fig. 6 CD31^hiEmcn^hi vessels and skeletal progenitors expand and infiltrate into subcritical-sized defects during healing. A–H** Blood vessels (**A–H**), Osterix+ progenitors (**A–D, A'–D'**), and Gli1+ progenitors (**E–H, E'–H'**) inside and around 1-mm parietal bone defects at PFD21 (**A, B, E, F**) and PFD56 (**C, D, G, H**). The dotted circle marks the defect region. **I, J** Expansion of Osterix+ and Gli1+ progenitors in the periosteum (yellow arrows) nearby the defect region (dotted lines) and sutures (yellow arrowheads) at PFD21. Progenitors were sparsely populated in the dura mater—the only portion of the calvarium that remained intact following defect injury. Results were replicated in 3 calvaria for each timepoint and staining combination (12 total calvaria). **K** Spatial relationship of skeletal progenitors to different vessel phenotypes at PFD21 and PFD56 ($n = 3$). **L–N** Fractional vessel phenotype volume (**L**), skeletal progenitor number (**M**), and total vessel volume (**N**) in the defect at PFD21 and PFD56 ($n = 6$ for **L**, **N**; $n = 3$ for **M**). **O** MicroCT quantification of defect to contralateral bone volume ratio at PFD21 and PFD56 ($n = 4$). Data are mean ± SD. Statistics were performed using a two-way ANOVA with Bonferroni post-hoc test (**K–M**) or two-tailed $t$-test (**N–O**). ***$p < 0.001$, **$p < 0.01$, *$p < 0.05$ where designated or between CD31^hiEmcn^hi and CD31^hiEmcn^− vessels in **K**; +++$p < 0.001$ between CD31^loEmcn^hi and CD31^hiEmcn^− vessels. Exact $p$-values for two-tailed $t$-tests: **N** $p = 0.0094$, **O** $p = 0.5660$. Scale bars: 500 μm (**I**, **J**); 300 μm (**A, C, E, G**); 200 μm (**B, D, F, H**). Colors: Red: Endomucin (Emcn), Gray: Osterix (**A–D, I, J**) or Gli1 (**E–H**), Green: CD31.

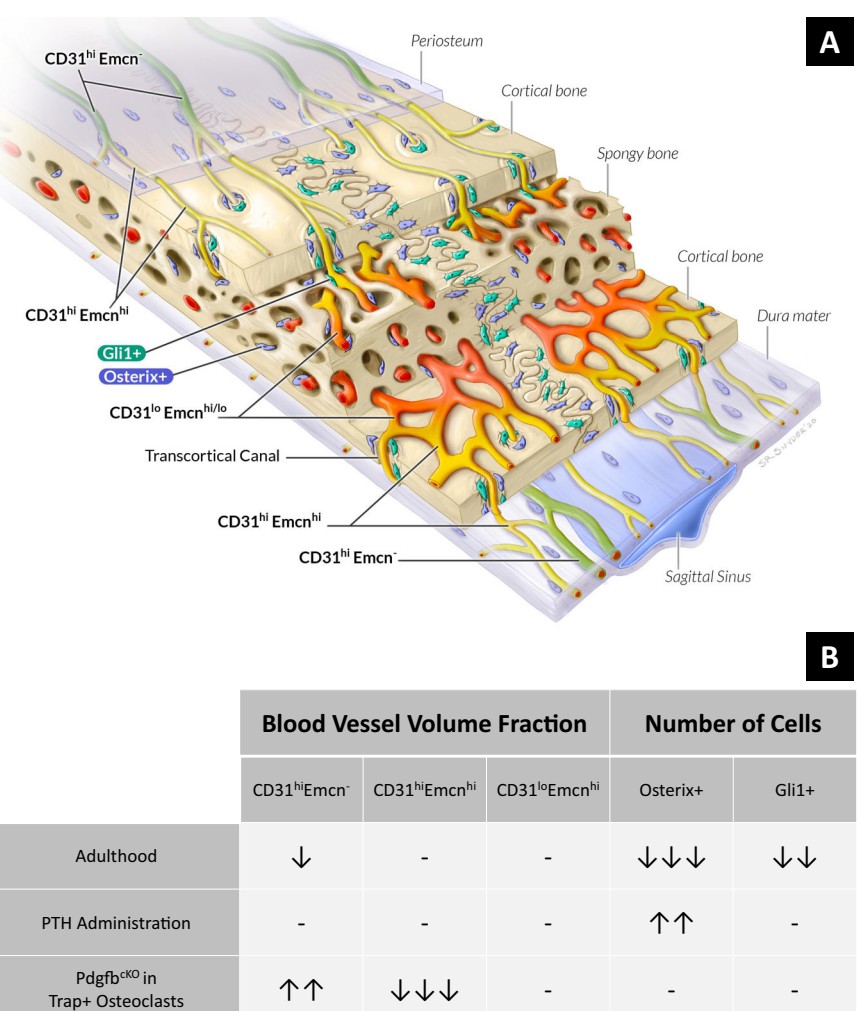

| | **Blood Vessel Volume Fraction** | | | **Number of Cells** | |
|---|---|---|---|---|---|
| | CD31$^{hi}$Emcn$^-$ | CD31$^{hi}$Emcn$^{hi}$ | CD31$^{lo}$Emcn$^{hi}$ | Osterix+ | Gli1+ |
| Adulthood | ↓ | - | - | ↓↓↓ | ↓↓ |
| PTH Administration | - | - | - | ↑↑ | - |
| Pdgfb$^{cKO}$ in Trap+ Osteoclasts | ↑↑ | ↓↓↓ | - | - | - |
| Healing (Early compared to late stage) | ↓↓ | ↑↑ | - | ↑↑ | ↑↑↑ |

**Fig. 7 Schematic representing the spatial distribution and prevalence of vessels and skeletal progenitors in the calvarium. A** Illustration showing the relative locations of the vessel and skeletal progenitor subtypes in the parietal bones from 4-weeks-old mice, as derived from Fig. 2. **B** Table demonstrating how the fractional volume of vessel subtypes and number of skeletal progenitors change during postnatal growth, alterations to remodeling, and healing.

healing, but it remains unclear how these relationships manifest in craniofacial bones. In this study, we provide a foundational framework for understanding how distinct vessel phenotypes and skeletal progenitors spatially interact in the calvarium (Fig. 7).

In long bone, recent studies have discovered distinct capillary subtypes—characterized by their expression of CD31 and endomucin (Emcn)—that exhibit divergent spatial relationships to active sites of bone growth and remodeling[15,24]. CD31$^{hi}$Emcn$^{hi}$ vessels are spatially associated with Osterix+ osteoprogenitors, provide signaling cues to support osteoprogenitors and perivascular cells, and are abundant in the primary spongiosa, periosteum, and endosteum of the long bone[15,21]. By contrast, CD31$^{lo}$Emcn$^{lo}$ vessels comprise the sinusoids of the diaphyseal bone marrow, where there are low numbers of osteoprogenitors[15]. CD31$^{hi}$Emcn$^{hi}$ vessels couple angiogenesis to osteogenesis in long bone through Notch signaling and secretion of pro-angiogenic factors and are necessary for maintaining bone mass in adulthood[24,25]. Further, it has been shown that a conditional knockout of PDGF-BB in TRAP+ osteoclasts reduces CD31$^{hi}$Emcn$^{hi}$ vessels in the periosteum and bone marrow and disrupts angiogenic–osteogenic coupling[21]. Collectively, these studies demonstrate the necessity of maintaining CD31$^{hi}$Emcn$^{hi}$ vessels to support proper growth and remodeling in long bone.

Our study sought to characterize the distribution of vessel subtypes in the calvarium and determine whether a similar preferential spatial relationship between CD31$^{hi}$Emcn$^{hi}$ vessels and skeletal progenitors exists in the cranial bone. First, we developed a quantitative 3D light-sheet imaging platform that illustrates the calvarial microvascular environment at a scale and resolution superior to other techniques. The most widely used imaging modalities used to study calvarial vasculature—immunohistochemistry and intravital microscopy—only allow for visualization of small regions and do not reflect the distinct 3D microenvironments in different regions of the calvarium. Recent advances in light-sheet imaging have enabled visualization of bone vasculature and cells over larger volumes, but these methods rely on endogenous fluorescence and do not allow for simultaneous visualization of multiple cellular markers[12,13]. Our light-sheet imaging platform overcomes these limitations by combining whole-mount immunostaining with an optical clearing reagent compatible with a wide range of antibodies and fluorophores—including endogenous fluorescent proteins[14,26,27]—and enables the study of spatial interactions between a variety of cell types. Even though our method requires long immunostaining incubations (2–3 weeks), our platform does not require decalcification

or complex clearing processes—both of which generally take at least 1 week to perform[12,28]. Furthermore, our semi-automated quantitative pipeline allows for high-throughput and consistent analysis of large data sets (>100 GB)—a challenging feat due to the advanced computational requirements for processing 3D light-sheet images. Our versatile imaging platform can be readily adapted to unveil other elusive biological phenomena in cranial bone biology that have been difficult to study with established techniques.

Using our imaging method, we present high-resolution 3D maps that illustrate the regional diversity in vessel phenotypes and skeletal progenitor populations. Prior to this report, studies have been limited to showing vessels and cells in small regions of the calvarium, and it was unclear how these structures are distributed through the entire volume. Our experiments demonstrate that most arteries and arterioles—high in CD31 and negative in Emcn expression[19]—are present in the periosteum and dura mater, while CD31$^{hi}$Emcn$^{hi}$ capillaries connect arterioles to bone marrow sinusoids via transcortical canals. Transcortical vessels and marrow sinusoids are primarily located near the sutures during adolescence, but they further develop toward the center of the frontal and parietal bones during cranial maturation. We also show that skeletal progenitors are concentrated at select regions near CD31$^{hi}$Emcn$^{hi}$ vessels—particularly at the transcortical canals and osteogenic fronts—and that Osterix+ osteoprogenitors are prevalent in the periosteal layers during active periods of bone growth. These regional differences reflect the importance of characterizing the microvascular environment throughout the calvarium as opposed to select areas of interest.

Our study is the first to demonstrate that Gli1+ skeletal stem cells exhibit an intimate spatial relationship with CD31$^{hi}$Emcn$^{hi}$ vessels in bone. Previous studies have shown that Gli1+ skeletal stem cells mainly reside in the calvarial sutures[16,29,30], and reported that they are not associated with vasculature[16]. However, our results suggest that a subset of these cells is associated with CD31$^{hi}$Emcn$^{hi}$ vasculature—particularly near the transcortical canals and marrow cavities, where there were also high concentrations of Osterix+ progenitors. Additionally, the Gli1 protein expression was brighter and more concentrated at the transcortical canals compared to the sutures. The discrepancy between this finding and earlier reports highlights the advantages intrinsic to using our established method to visualize these cells: Prior studies primarily used transgenic reporter mice to label skeletal stem cells—a method less sensitive to differences in protein expression levels compared to antibody staining[31]. Additionally, these studies used thin tissue sections to characterize Gli1+ cell distribution, making it difficult to capture the transcortical canals in calvarial bone[32].

Our results also demonstrate key differences in the relationship between Osterix+ osteoprogenitors and CD31$^{hi}$Emcn$^{hi}$ vessels in the calvarium compared to the long bone. While we found that Osterix+ osteoprogenitors are preferentially associated with CD31$^{hi}$Emcn$^{hi}$ vessels, the volume of these vessels was not directly correlated with the total number of Osterix+ cells. In our experiments evaluating postnatal growth, we found that Osterix+ cell number was higher in the calvaria of juvenile mice compared to adult mice, yet CD31$^{hi}$Emcn$^{hi}$ volume remained the same in both groups. Similarly, when exploring stimulated remodeling via PTH administration, we found that the number of Osterix+ cell was greater in PTH-treated versus non-PTH-treated mice, without concomitant changes in the volume of CD31$^{hi}$Emcn$^{hi}$ vessels. These findings differ from those observed in long bone, where increases of CD31$^{hi}$Emcn$^{hi}$ vessels lead to higher Osterix+ osteoprogenitor numbers and bone mineral deposition[33,34]. The distinct niches that are present in long bone versus the calvarium

may account for this discrepancy. In instances when Osterix+ cells were elevated in the calvarium, much of the increase occurred in the periosteum and dura mater, where CD31$^{hi}$Emcn$^{hi}$ vessels appeared to have a less direct spatial relationship with osteoprogenitors. By contrast, most studies in long bone have evaluated vessels in the metaphyseal region adjacent to the growth plates—an anatomical feature that is not present in calvarial bone. These differences demonstrate the necessity in studying angiogenic-osteogenic coupling in cranial bone separately from other bone types.

Additionally, we found that osteoclast signaling may be important for maintaining the spatial relationship of CD31$^{hi}$Emcn$^{hi}$ vessels and skeletal progenitors in the calvarium. When we employed a mouse model where PDGF-BB is conditionally knocked out in TRAP+ osteoclasts, the percentage of Osterix+ and Gli1+ progenitors in close proximity (<5 μm) to CD31$^{hi}$Emcn$^{hi}$ vessels significantly decreased despite the total number of each cell type remaining the same. These data are similar to trends observed in long bone, where the loss of TRAP+ cell-secreted PDGF-BB reduces the number of CD31$^{hi}$Emcn$^{hi}$ vessels and migration of periosteal skeletal progenitors to the cortical bone surface[21,35]. In the calvarium, loss of this spatial coupling was most apparent at the transcortical canals. Other studies have shown that osteoclasts are important for maintaining the structure and function of these canals as channels for vascular-mediated nutrient and immune cell transport[32,36]. Furthermore, a recent study has demonstrated that vessel-associated osteoclasts at the primary spongiosa are critical for maintaining CD31$^{hi}$Emcn$^{hi}$ vessel homeostasis in the metaphysis[23]. Given the strong association of osteoclasts to CD31$^{hi}$Emcn$^{hi}$ vessels in the bone marrow and transcortical canals, our experimental observations suggest that osteoclasts support a regenerative niche for cranial bone growth and remodeling.

With regards to bone healing, our study provides insights into the contributions of vessel phenotypes and skeletal progenitors to cranial bone regeneration. While studies have documented spatial relationships between blood vessels and osteoblasts during calvarial healing[37,38], it has remained unclear whether specific vessel phenotypes are involved with this process. We found that the majority of vessels present at the early stages of healing (PFD21) were high in CD31 and Emcn expression—the stage at which most bone mineral deposition also took place. Furthermore, CD31$^{hi}$Emcn$^{hi}$ vessels were most associated with Gli1+ and Osterix+ cells within the defects at early and later stages of healing, despite the fact that CD31$^{hi}$Emcn$^{hi}$ vessels had regressed at PFD56. Provided that little bone formed between PFD21 and PFD56, our data suggest that CD31$^{hi}$Emcn$^{hi}$ vessels may be critical in driving calvarial defect healing.

We also found that Gli1+ cells systemically expanded following injury—an observation that has not been previously documented. Other studies have suggested that cranial skeletal stem cells—including Gli1+ cells—migrate from the sutures to the surrounding bone following injury[16,39,40]. Furthermore, studies have shown that stem cells in the periosteum expand and contribute to calvarial defect healing[41,42]. However, these studies only evaluated regions near the site of injury, and it was unknown whether skeletal progenitors expanded in non-injured areas of the calvarium. In our experiments, Gli1+ cells proliferated in the periosteum over and around the defect site, as well as nearby the sagittal and coronal sutures. At PFD56, we observed excess mineral deposits along the superficial cortical surface of the uninjured bone, suggesting that this systemic response contributed to bone mineral deposition outside of the defect region. These observations provide a rationale for future studies to evaluate the effects of this systemic healing response to determine how it impacts calvarial structure and function.

Our work provides an essential foundation for studying calvarial bone that will enable others to build upon our findings and progress the field of craniofacial bone biology. While we have provided detailed insight into the spatial coupling of skeletal progenitors to vessel phenotypes, there still remains significant knowledge gaps in how other cell types, such as neurons and immune cells, contribute to angiogenic-osteogenic coupling during calvarial growth, healing, and remodeling. Our versatile and inexpensive platform can be readily adapted to observe these different cell types and systematically study a variety of biological processes in the calvarium. Harnessing the powerful capabilities of our quantitative 3D imaging approach will broaden our understanding of craniofacial bone biology and accelerate the development of effective treatments for debilitating craniofacial bone injuries and disorders.

## Methods

**Materials**. All essential antibodies, reagents, animal drugs and materials, instruments and hardware, and software used in this study are provided in Supplementary Table 1.

**Study approval**. All animal experiments were approved by the Johns Hopkins University Institutional Animal Care and Use Committee (Protocol No. MO18M188). Animals were housed and cared for in Johns Hopkins' Research Animal Resources central housing facilities.

**Experimental animals**. We purchased the following mouse strains from Jackson Laboratories: C57BL/6J (Stock No. 000664) and *Pdgfb*[fl/fl] (Stock No. 017622). We obtained *Trap-cre* mice from J.J. Windle (Virginia Commonwealth University, Richmond, VA, USA). *Trap-cre Pdgfb*[fl/fl] mice were generated using a previously published protocol[21]. Briefly, hemizygous *Trap-cre* mice were crossed with *Pdgfb*[fl/fl] mice to produce *Trap-cre Pdgfb*[fl/fl] offspring (referred to as Pdgfb[cKO] in the Main section). *Pdgfb*[fl/fl] mice were used as a control and referred to as WT in the Main section. Mice genotype was confirmed by performing PCR on DNA isolated from mouse toes using primers designated previously[21].

**Murine calvarial harvest**. To harvest calvaria, we perfused the vasculature with heparinized saline (10 U/mL in 1X PBS) to remove the blood in calvarial bone prior to fixation. Mice were heavily anesthetized with ketamine (100 mg/kg) and xylazine (20 mg/kg) and subcutaneously injected with 200 U heparin to prevent premature clotting. An initial incision was made near the xiphoid process, and the chest was then cut open along the lateral edges of the rib cage to provide access to the heart. Heparinized saline was perfused into the left ventricle via a blunt 20 G needle at a rate of 10 mL/min. The right atrium was opened just prior to perfusion to enable open circulation. Following perfusion, calvaria were harvested—taking special care to preserve the periosteum and dura mater—and were fixed in 4% methanol-free paraformaldehyde overnight at 4 °C. Fixed calvaria were washed with PBS three times prior to staining.

**PTH administration**. We administered PTH daily for 1 month to determine the effects of stimulated bone remodeling on calvarial vessels and skeletal progenitors. 40 μg/kg pTH (1–34) was injected subcutaneously into male 8-weeks-old C57BL/6J mice 5 days/week for 4 weeks. Mice were harvested at 12 weeks of age on the day following the last PTH dose. 12-weeks-old male C57BL/6J mice without PTH treatment were used as the control for quantitative comparisons.

**Calvarial defect procedure**. We created a subcritical 1-mm sized defect in the parietal bone to characterize the response by vessel phenotypes and skeletal progenitors during calvarial healing. 8-weeks-old male C57BL/6J mice were weighed and anesthetized via a single intraperitoneal injection using ketamine (100 mg/kg) and xylazine (10 mg/kg). The paw-pitch test was performed to determine the level of sedation. Then, mice were placed on a stereotaxic frame to fix the head. Buprenorphine SR (1 mg/kg) was injected subcutaneously to provide sustained analgesia following the procedure. To access the parietal bone, the skin was shaved, treated with alcohol and betadine, and draped under sterile conditions. Sterile gloves and masks were used by all surgical personnel, and all the surgical procedures were performed under an operating microscope by a single surgeon. A mid-sagittal incision was made over the center of calvarium. Following identification of anatomical landmarks, a 1-mm defect was created ~1–2 mm away from the sagittal suture using a microsurgical drill and 1 mm carbide inverted bone burr. Special precaution was taken to preserve the underlying dura mater. Following defect creation, the skin incision was closed using 6-0 nylon monofilament sutures. Mice were monitored daily up to 1 week following surgery for any neurological deficit, infection, pain, or distress. After 21- and 56-days following surgery, all mice were euthanized and calvaria were harvested according to the same procedure described above.

**Whole-mount immunostaining and optical clearing**. To enable 3D light-sheet imaging of calvaria, we performed whole-mount immunostaining to label blood vessel phenotypes and skeletal progenitors and optically cleared the calvaria following staining. First, samples were blocked overnight at 4 °C using a solution comprised of 10% V/V normal donkey serum in wash buffer (0.1 M Tris, 0.15 M NaCl, 0.05% V/V Tween-20, 20% V/V dimethylsulfoxide, pH 7.5), and then using a biotin blocking kit for 8 h at room temperature to mask endogenous biotin. Samples were stained with primary antibodies for CD31, Emcn, and Osterix or Gli1 for 7 days, fluorophore- and biotin-conjugated secondary antibodies for 7 days, and a streptavidin conjugate for 5 days to enable signal amplification of Emcn. All antibodies and conjugates were diluted in the same buffer used for blocking. Calvaria were washed five times over a 24 h period between antibody and streptavidin incubation steps. Following staining, samples were cleared using a graded series of 2,2-thiodiethanol (TDE in TBS-Tween; 25%, 50%, 75%, 90%, 100% × 2). Each clearing step was performed for 2 h at room temperature or overnight at 4 °C. Calvaria were stored in 100% TDE at 4 °C prior to imaging.

**Light-sheet imaging**. We imaged calvaria using a custom light-sheet imaging protocol that enabled us to achieve adequate signal intensity and quality throughout the entire volume. Calvaria were mounted and immersed into a glass imaging chamber containing 100% TDE. The chamber was loaded into a LaVision Biotec Ultramicroscope II that was pre-aligned to match the refractive index of TDE. Whole calvarial samples were imaged using three separate acquisitions: (1) a 3 × 1 tile using double-sided illumination at the center of the sample, (2) a 3 × 2 tile using left-sided illumination at the left portion of the sample, and (3) a 3 × 2 tile using right-sided illumination at the right portion of the sample. Calvaria with defects were imaged in a single acquisition using a 2 × 2 tile region and single-sided illumination. Tiles were overlapped by 15% within each acquisition and 30–35% along the x axis between different acquisitions to facilitate stitching. The following hardware and settings were used for all scans: ×2.5 zoom with a ×2 dipping cap (×5 magnification, 1.3 μm x–y pixel size), 5.5 Megapixel sCMOS camera, 20 ms exposure time, 0.154 numerical aperture, and 2.5 μm z step size. Based upon the assumption that the light-sheet followed a Gaussian beam profile, the estimated light-sheet width was 2.3–3.2 μm at the center of each tile and 284.2 μm at the horizontal edges of each tile for all scans[43,44]. Different channels were imaged using 561, 640, and 785 nm lasers and 620/60, 680/30, and 845/55 filters, respectively. Laser powers were optimized for each antibody and held constant between scans.

**Image processing and analysis**. We performed all image processing and analysis using Imaris 9.5 software and a Dell Precision 7820 Tower workstation. The workstation was equipped with a Dual Intel Xeon Gold 6240 processor, 384 GB DDR4 SDRAM (2666 MHz speed), 512 GB and 1 TB SATA SSDs, NVIDIA Quadro RTX5000 graphics card (16 GB GDDR6 memory), and Windows 10 Pro for Workstations. All data were stored and analyzed using Samsung T5 2TB External SSDs connected via USB 3.2 or USB 3.2 Type-C ports.

To pre-process the images for analysis, we converted LaVision Biotec raw OME-TIFF files to the Imaris file format (.ims) for each individual tile using Imaris File Converter 9.5. Tiles were manually aligned along the x–y axes and stitched into one 3D image using Imaris Stitcher 9.5.

Following image pre-processing, we implemented a custom analysis pipeline in Imaris 9.5 to enable us to characterize vessel phenotypes and skeletal progenitors. First, a pre-defined VOI containing six distinct rectangular regions was positioned for each dataset, with the VOI dimensions remaining constant across all datasets. The sagittal and interfrontal sutures were avoided due to their lower signal quality. Following VOI specification, CD31[hi] and Emcn[hi] vessels were segmented using the Surfaces module with a 10 μm radius for background subtraction and $10^4$ μm$^3$ volume filter to eliminate subcellular-sized segments. Osterix+ and Gli1+ cells were segmented using the Spots module using a pre-measured spot size in the axial and lateral dimensions (5 μm lateral, 15 μm axial for Gli1+; 6 μm lateral, 18 μm axial for Osterix+ cells). Thresholds were optimized for each experimental group to minimize the background in the segmented objects. Following this initial segmentation, images were down-sampled by a factor of two in each dimension to facilitate a second round of segmentation for vessel phenotypes. In the down-sampled data sets, binary masks for CD31- and Emcn-segmented vessels were created and re-segmented using the "Split Objects" Surfaces function (10 μm seeding point diameter). CD31[hi]Emcn− and CD31[hi]Emcn[hi] vessels were segmented using the CD31 mask and filtered based upon the absence or presence of masked Emcn signal within each object, respectively. CD31[lo]Emcn[hi] vessels were segmented based upon the Emcn mask and filtered to remove objects co-localized with masked CD31 signal. During all segmentation steps, "shortest distance calculation" was activated to enable measurements of individual cells to their nearest vessel. Cells touching their nearest vessel were designated as 0 μm away from the vessel.

To quantify vasculature and skeletal progenitors in our defect model, we applied the same approach using modified VOIs. A VOI with a 1 mm × 1 mm x–y area was placed in the defect region to calculate vessel volume and skeletal progenitor

number within the defects. Additionally, to assess the systemic response to injury, an expanded VOI with a 2.3 mm × 3.6 mm *x–y* area was positioned in the ipsilateral and contralateral parietal bone approximately 1 mm away from the sagittal suture. The z-dimension of all VOIs spanned the full thickness of the calvarial bone.

Once all image segmentation was complete, we exported Surface and Spots statistics for vessel phenotypes and skeletal progenitors, respectively, to enable data analysis in third party software. XiT software[14] was used to plot the spatial distance of individual Osterix+ or Gli1+ cells relative to each vessel phenotype in 3D. GraphPad Prism and Microsoft Excel were used to plot and analyze vessel volume, cell number, and vessel-cell distance measurements.

**MicroCT scanning**. We used µCT to analyze changes in bone microstructure between different experimental groups. Prior to scanning, TDE was gradually removed from the calvaria by washing in 50% TDE and then PBS several times at room temperature. Calvaria were scanned using a Bruker Skyscan 1275 µCT with a 1 mm aluminum filter, 65 keV source voltage, 0.3° step rotation, and 9 µm voxel size.

To analyze calvarial microstructure, we implemented a custom analysis pipeline in CTAN and CTVOL software. Scans were re-sliced along the transverse plane to provide a uniform scan orientation among all data sets. In the re-sliced scans, a rectangular VOI was selected at the mid-point of the parietal and posterior frontal bones with dimensions of 4.95 mm × 3.6 mm × 0.9 mm and 3.6 mm × 3.6 mm × 0.45 mm, respectively (first two dimensions along transverse plane; latter two dimensions along sagittal plane). For *Trap-cre Pdgfb^{fl/fl}* mice and their WT littermates, the VOI dimensions were reduced to 4.05 mm × 2.7 mm × 0.9 mm in the parietal bone and 2.7 mm × 2.7 mm × 0.45 mm in the posterior frontal bone. Calvarial bone was segmented using a pre-defined threshold that remained constant across all scans. The shrink wrap function was subsequently performed to reduce the VOI to the boundaries of the tissue volume. 3D analysis was performed on the final segmented structure to assess bone volume (BV), bone volume/tissue volume (BV/TV), and bone surface area to volume ratio (SA/V).

For our calvarial defect experiments, we quantified newly formed bone in the defects using Mimics 14 software. Bone was segmented using a pre-defined threshold that was held constant among all datasets. Cylindrical VOIs with a 1 mm diameter were placed within the defect and in the contralateral parietal bone to allow for quantification of defect/contralateral bone volume.

**Statistics**. We used GraphPad Prism 5 software to perform all statistical analyses. All measurements were performed on distinct samples for each type of analysis. Statistical comparisons were performed using a two-tailed *t*-test, one-way ANOVA with Tukey's post-hoc test, or two-way ANOVA with Bonferroni's post-hoc test. Statistical tests performed and sample sizes for each dataset is designated in the figure captions. All *p*-values <0.05 were considered statistically significant.

**Reporting summary**. Further information on research design is available in the Nature Research Reporting Summary linked to this article.

## Data availability

The data supporting the findings from this study are available within the article file and its supplementary information. Source data are provided with this paper. The 3D microscopy data generated in this study have been deposited in the BioImage Archive database under accession code S-BIAD171. Any remaining raw data will be available from the corresponding author upon reasonable request. Source data are provided with this paper.

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

## Acknowledgements

We are appreciative of several individuals for their assistance with this study: Yuan Cai (light-sheet microscope training and alignment), Dr. Scot Kuo and Dr. Hoku West-Foyle (image software training and general advice), Jasmine Hu (staining protocol training), Shawna R. Synder (schematic illustration in Fig. 7a), and Dr. Jennifer Elisseeff (review of the manuscript). We also thank the JHU Integrated Imaging Center and JHU SOM Microscope Facility for providing the equipment and software necessary to carry out this work. This work was supported by the National Science Foundation Graduate Research Fellowship (A.N.R.), NIH National Institute for Dental and Craniofacial Research Grant No. 5 F31 DE029109-02 (A.N.R.) and 5 R01 DE027957-02 (W.L.G.), ARCS Foundation Metropolitan Washington Chapter (A.N.R.), and NIH Shared Instrumentation Grant No. 1S10OD020152-01A1 (Integrated Imaging Center).

## Author contributions

A.N.R., M.W. and W.L.G. conceived the study. A.N.R. performed all mouse harvests, PTH injections, staining, light-sheet imaging, data analysis, and manuscript writing. X.L. performed the microCT scans and bred the *Trap-cre Pdgfb^{fl/fl}* mice. S.F.H. and D.L.C. helped design the staining and image analysis protocols. A.P.-P. and A.N.R. performed the calvarial defect procedures. M.W., T.F.W. and W.L.G. provided key materials and support. All authors reviewed the manuscript and discussed the work.

## Competing interests

The authors declare no competing interests.
