## [Peer Review File · Nature Communications]

REVIEWER COMMENTS

Reviewer #1 (Remarks to the Author):

The work by Rindone and colleagues describes the development of a quantitative 3-D imaging platform enabling to visualize and analyze high-resolution data set of whole murine calvarium.

This technique provides a powerful tool for the single-cell resolution 3-D map of vessels and skeletal progenitors cells within the frontal and parietal cranial bones.

By employing this novel platform, the authors unveil a spatial relationship between CD31+/Emnc+ vessels and Osterix+ (Osx+) and Gli+ skeletal progenitors during the calvarial growth, bone remodeling and regeneration. Moreover, the use of this platform defines the microvascular environment of the skull bones at high resolution.

The study represents a valuable technical accomplishment and although descriptive introduces a substantial novelty and great opportunity for a large number of applications in the field of craniofacial biology and medicine research. Overall the study is relevant and provides an interesting foundation for future studies aimed at elucidating bone development and regeneration.

Comments:

1. Figure 1, Panel A shows presence of Osx+ skeletal progenitors in several areas of both frontal and parietal bones, while Osx+ cells are not visible in the cranial PF and SAG sutures. Could the authors comment on this? Furthermore, it is not clear whether the opalescent white staining within the COR suture is representative of Osterix+ cells.
2. How did the authors assess for the High and Low levels of CD31 and Emnc? Do they evaluate the intensity of staining? In that case, how quantitative and reliable is this technique?
3. In Figure 3, the authors compare the distribution of Osx+ and Gli+ skeletal osteoprogenitors between Juvenile (4-wks) and adult (12-wks) mice showing (as would be expected) a decline of this population distribution in adult mice. Have the authors analyzed in depth whether regional differences in Osx+ and Gli+ cells distribution occur in frontal and parietal bones? As well as their spatial relationship with vessels?

4. Could the authors comment/discuss on the preferential association of both Osx+ and Gli+ cells with CD31^{high} Emnc^{high} rather than with CD31^{high} Emnc^{low} or CD31^{low} Emnc^{low} vessel phenotype?

5. Have the authors applied their platform to study other craniofacial bones, for example the mandible? Is this method suitable for other craniofacial bone?

Reviewer #2 (Remarks to the Author):

Regarding imaging, the manuscript is ok. The authors just used a standard light sheet microscope in their imaging facility and obtained the results that are possible with this instrument. As far as I understand, the authors also do not imply that regarding imaging anything would be novel.

Regarding clearing I am astonished that just immersion in TDE was sufficient but that obviously helped to get quite impressive stainings. The z resolution of the Ultramicroscope in this macroscopic regime is rather poor but the xy resolution is fine and allowed the authors to show nice recordings. I would like to see some quantitative numbers how thick the light sheets were in the different images, both in the center and at the sides of the tiles used. As the authors finally show a 3D reconstruction I suggest that the authors use at least on pieces of their preparation a microscope with higher z resolution to substantiate their findings in axial direction.

If results with better axial resolution for the sensitive statements of their findings can be shown, from the imaging side the article can be published.

Reviewer #3 (Remarks to the Author):

Vascularization is critical for the development and growth of craniofacial bone. To understand the relationship between blood vessels and osteoprogenitors during craniofacial bone growth and remodeling, Rindone et al developed a quantitative 3D imaging platform to visualize the spatial distribution of blood vessels and osteoprogenitors throughout the entire murine calvarium. This is an interesting study, and the method is broad of interest. However, this is an observational study, and following limitations should be addressed before publication.

- 1) The imaging method requires sacrificing the animals, which is not longitudinal, and is not suitable for the observation of the dynamic biological process.
- 2) This method requires staining with specific antibodies, which normally allow for using three different fluorophores. This also limits its application.
- 3) It takes long time (about 3 weeks) to immunostaining tissues using this method, also limits its application.
- 4) The imaging method is not easy applied for visualizing other tissues due to its limited depth.
- 5) The study explored the spatial relationship of blood vessels and skeletal progenitors in the calvarium using immunostaining of CD31 and EMCN. It would be interesting to know the relationship of skeletal progenitors in the artery, capillary, and venues.
- 6) Osteoprogenitors distributed differentially in the blood vessels. During healing and remodeling, how skeletal progenitors migrate along the blood vessels and how they contribute to bone regeneration?
- 7) Osteoprogenitors are heterogeneous. It would be interesting to know the spatial relationship of different subtypes of osteoprogenitors and blood vessels. Further study to reveal how different subtypes of osteoprogenitors contribute to bone remodeling.

REVIEWER

COMMENTS

The authors would like to thank the reviewers for their critical review of the manuscript and for their helpful comments. We have provided a point-by-point response below and have revised the manuscript based on these suggestions. We believe these changes strengthen the manuscript. Major changes are highlighted in red font in the revised manuscript.

Reviewer #1 (Remarks to the Author):

The work by Rindone and colleagues describes the development of a quantitative 3-D imaging platform enabling to visualize and analyze high-resolution data set of whole murine calvarium.

This technique provides a powerful tool for the single-cell resolution 3-D map of vessels and skeletal progenitors cells within the frontal and parietal cranial bones.

By employing this novel platform, the authors unveil a spatial relationship between CD31+/Emnc+ vessels and Osterix+ (Osx+) and Gli+ skeletal progenitors during the calvarial growth, bone remodeling and regeneration. Moreover, the use of this platform defines the microvascular environment of the skull bones at high resolution.

The study represents a valuable technical accomplishment and although descriptive introduces a substantial novelty and great opportunity for a large number of applications in the field of craniofacial biology and medicine research. Overall the study is relevant and provides an interesting foundation for future studies aimed at elucidating bone development and regeneration.

We thank the reviewer for the positive comments. We have provided responses to each individual comment below.

1. Figure 1, Panel A shows presence of Osx+ skeletal progenitors in several areas of both frontal and parietal bones, while Osx+ cells are not visible in the cranial PF and SAG sutures. Could the authors comment on this? Furthermore, it is not clear whether the opalescent white staining within the COR suture is representative of Osterix+ cells.

In Figure 1, Panel A, the green CD31 signal from the sagittal sinus overwhelms the signal from the Osterix+ cells. However, in Figure 2A, where we show the individual channel, Osterix+ cells can be clearly seen in these suture regions (including individual Osterix+ cells in the coronal suture).

2. How did the authors assess for the High and Low levels of CD31 and Emnc? Do they evaluate the intensity of staining? In that case, how quantitative and reliable is this technique?

We clarify the description of this process in Paragraph 3 of the "Image Processing and Analysis" Methods section. Constant fluorescent thresholds for each experimental group were set to maximize the number of segmented vessels while minimizing the background signal (defined as regions visibly outside of the CD31 and Emcn fluorescence signals). Signal below this threshold was considered "low" CD31 and Emcn expression since some non-segmented CD31^{lo}Emcn^{lo} sinusoids were present in the

marrow cavities. Beyond this initial thresholding step, we did not evaluate the intensity of staining for any of the markers. Using this approach, we found that our segmentation results were consistent between samples.

3. In Figure 3, the authors compare the distribution of *Osx*⁺ and *Gli*⁺ skeletal osteoprogenitors between Juvenile (4-wks) and adult (12-wks) mice showing (as would be expected) a decline of this population distribution in adult mice. Have the authors analyzed in depth whether regional differences in *Osx*⁺ and *Gli*⁺ cells distribution occur in frontal and parietal bones? As well as their spatial relationship with vessels?

We thank the reviewer for this suggestion. We have performed additional analyses of the parietal bones, coronal suture (including frontal/parietal bones just adjacent to the suture), and frontal bones. These data are presented in a new Supplementary Figure 4. While all regions showed declines in skeletal progenitor numbers between 4-wk and 12-wk old mice, we found that the most significant decreases occurred in the parietal bone. The preferential spatial relationships of skeletal progenitors and *CD31*^{hi}*Emcn*^{hi} vessels were maintained across all regions at both ages. This relationship was moderately weaker in the coronal suture region compared to the parietal and frontal bones. To reflect these data, we have made changes to the “Post-natal growth shifts the distribution of vessel phenotypes and *Osterix*⁺ progenitors” subsection of the Results:

*“Along with changes in vessel phenotypes, the numbers of *Osterix*⁺ and *Gli*⁺ skeletal progenitors decreased in adult calvaria, and their distribution was mainly restricted to sutures, transcortical canals, and bone marrow cavities (Fig. 3A-F, G). While a decrease in progenitors was observed in different regions of the calvarium, the most significant decline occurred in the parietal bones (Supp. Fig. 4A). Moreover, *Osterix*⁺ cells were mostly absent in the periosteum and dura mater of adult calvaria (Fig. 3A-C). These results correlated with differences in vessel-progenitor relationships: The fraction of *Osterix*⁺ cells within 5 μm of the nearest vessel in adult versus juvenile calvaria was significantly higher for *CD31*^{hi}*Emcn*^{hi} and *CD31*^{lo}*Emcn*^{hi} vessels and lower for *CD31*^{hi}*Emcn*⁻ vessels (Fig. 3J). These trends held across different regions of the calvarium (Supp. Fig. 4B). There were no significant changes in the relationship of *Gli*⁺ cells to vessel phenotype between juvenile and adult calvaria (Fig. 3J). Nonetheless, both progenitor cell types maintained a preferential spatial association with *CD31*^{hi}*Emcn*^{hi} vessels at 4 and 12 weeks of age.”*

4. Could the authors comment/discuss on the preferential association of both *Osx*⁺ and *Gli*⁺ cells with *CD31*^{high} *Emcn*^{high} rather than with *CD31*^{high} *Emcn*^{low} or *CD31*^{low} *Emcn*^{low} vessel phenotype?

We now address this in Paragraph 2 of the discussion. Prior studies have described a preferential association of *Osterix*⁺ cells with *CD31*^{hi}*Emcn*^{hi} vessels in the primary spongiosa of long bones and demonstrated that these vessels express signaling cues that support osteoprogenitors and perivascular cells (references: Kusumbe AP et al., Nature 507, 323-328(2014); Kusumbe AP et al., Nature 532, 380-384(2016)). While we did not evaluate the mechanism behind these spatial relationships in the calvarium, based upon results observed in long bone, we would hypothesize that these vessels express

molecules supportive of skeletal stem cells and osteoprogenitors, but this would need to be tested in a separate study.

5. Have the authors applied their platform to study other craniofacial bones, for example the mandible? Is this method suitable for other craniofacial bone?

We have not yet applied the platform to study other craniofacial bones, but we plan to adapt it to other applications in future studies. We anticipate that this method – with some modifications (i.e. decolorizing, delipidating) to account for the increased marrow volume – will be suitable for other bones such as the mandible.

Reviewer #2 (Remarks to the Author):

Regarding imaging, the manuscript is ok. The authors just used a standard light sheet microscope in their imaging facility and obtained the results that are possible with this instrument. As far as I understand, the authors also do not imply that regarding imaging anything would be novel.

Regarding clearing I am astonished that just immersion in TDE was sufficient but that obviously helped to get quite impressive stainings.

We thank the reviewer for their comments.

The z resolution of the Ultramicroscope in this macroscopic regime is rather poor but the xy resolution is fine and allowed the authors to show nice recordings. I would like to see some quantitative numbers how thick the light sheets were in the different images, both in the center and at the sides of the tiles used.

We used a light-sheet with a 5 μm thickness (2.5 μm z-step, 50% overlap between slices) and 5x magnification for all images in this study. We do not know the quantitative difference between the sheet thickness at the center versus the edges of a tile. However, we have included a representative image of a tile in the lateral and axial directions below (MIP of entire tile in the lateral direction, 80 μm -thick MIP in the axial direction). Additionally, we aligned the light-sheet weekly to ensure that the optics matched the refractive index of TDE. We found that this approach was sufficient to distinguish individual cells in our samples.

Representative tile image (scale bar 400 μm):

As the authors finally show a 3D reconstruction I suggest that the authors use at least on pieces of their preparation a microscope with higher z resolution to substantiate their findings in axial direction. If results with better axial resolution for the sensitive statements of their findings can be shown, from the imaging side the article can be published.

As a point of comparison, we have also included orthogonal images (80 μm -thick) taken from a confocal microscope and our light-sheet microscope below (scale bars 500 μm for zoomed-out, 100 μm for zoomed-in). Individual cells can be visualized using both imaging methods.

Note: Optical sections are not identical because they are from different samples and were imaged at slightly different orientations.

Reviewer #3 (Remarks to the Author):

Vascularization is critical for the development and growth of craniofacial bone. To understand the relationship between blood vessels and osteoprogenitors during craniofacial bone growth and remodeling, Rindone et al developed a quantitative 3D imaging platform to visualize the spatial distribution of blood vessels and osteoprogenitors throughout the entire murine calvarium. This is an interesting study, and the method is broad of interest. However, this is an observational study, and following limitations should be addressed before publication.

We thank the reviewer for their comments and suggestions. Our responses to each comment are included below.

1) The imaging method requires sacrificing the animals, which is not longitudinal, and is not suitable for the observation of the dynamic biological process.

The reviewer is correct. Our method is focused on observing the spatial distribution of cellular structures throughout the entire calvarium and is not intended to observe dynamic processes. While this is a limitation, it also provides several advantages and can supplement dynamic images obtained from different platforms such as those presented in our earlier studies (Mendez A, Rindone AN, et al., TISS. Eng. Pt C 24, 430-440(2018)).

2) This method requires staining with specific antibodies, which normally allow for using three different fluorophores. This also limits its application.

Our method focused on using antibody staining because it allowed us to stain up to three markers simultaneously. This is a limitation but also provides some advantages. For example, this antibody-based approach allowed us to identify the spatial relationships between Gli1+ cells and the vasculature, which was not previously described. In addition, for applications requiring a smaller field of view, a larger number of fluorophores may be used with confocal imaging (Coutu DL, et al., Nat. Biotech. 35, 1202-1210(2017)). Finally, we anticipate that our clearing method would be compatible with endogenous fluorophores. We added clarification on this in Paragraph 3 of our Discussion section:

“Our light-sheet imaging platform overcomes these limitations by combining whole-mount immunostaining with an optical clearing reagent compatible with a wide range of antibodies and fluorophores—including endogenous fluorescent proteins^{14,26,27}—and enables the study of spatial interactions between a variety of cell types.”

3) It takes long time (about 3 weeks) to immunostaining tissues using this method, also limits its application.

Again, the reviewer is correct. While the long immunostaining time is a limitation, we found that increasing the staining time enabled better antibody penetration into the tissue. However, our method saves time over other optical clearing approaches since it does not require decalcification (usually takes

1-2 weeks) or complex clearing processes (usually takes >1 week). We have added the following sentence to Paragraph 3 of the Discussion section to address this limitation:

“Even though our method requires long immunostaining incubations (2-3 weeks), our platform does not require decalcification or complex clearing processes—both of which generally take at least one week to perform^{12,28}.”

4) The imaging method is not easily applied for visualizing other tissues due to its limited depth.

Since our study was specifically focused on the calvarium, we have not yet applied the platform to thicker bones or other tissues. Our method may be suitable for other tissues when combined with other steps such as lipid removal and decolorization.

5) The study explored the spatial relationship of blood vessels and skeletal progenitors in the calvarium using immunostaining of CD31 and EMCN. It would be interesting to know the relationship of skeletal progenitors in the artery, capillary, and venues.

We have defined these different types of vessels based upon their levels of CD31 and Emcn expression (based upon knowledge from prior literature), as described in the second paragraph of the Results section:

“We applied the spots and surfaces modules in Imaris software to segment blood vessels and skeletal progenitors, and then performed a series of masking and filtering algorithms to denote three vessel phenotypes: CD31^{hi}Emcn⁻ arteries and arterioles, CD31^{hi}Emcn^{hi} capillaries, and CD31^{lo}Emcn^{hi} capillaries and sinusoids^{18,19}.”

We also discuss the distribution of these vessels and their spatial relationships to skeletal progenitors in the fourth paragraph of the Discussion section.

6) Osteoprogenitors distributed differentially in the blood vessels. During healing and remodeling, how skeletal progenitors migrate along the blood vessels and how they contribute to bone regeneration?

We did not investigate the mechanisms of skeletal progenitor migration and their precise contributions to bone regeneration in this study, as we focused on developing the imaging and quantitation pipelines. However, the methods can be applied to study these in future studies.

7) Osteoprogenitors are heterogeneous. It would be interesting to know the spatial relationship of different subtypes of osteoprogenitors and blood vessels. Further study to reveal how different subtypes of osteoprogenitors contribute to bone remodeling.

We confirm that we did evaluate two different sub-populations of skeletal progenitors to account for heterogeneity. Gli1 represents skeletal stem cells (less mature skeletal progenitors) while Osterix represents progenitors restricted to the osteoblast lineage. We have added the following clarification in the first paragraph of the Results section:

“Osterix is a marker for skeletal progenitors that are restricted to the osteoblast lineage¹⁵, while Gli1 marks less-differentiated skeletal stem cells^{16,17}.”

REVIEWERS' COMMENTS

Reviewer #1 (Remarks to the Author):

The authors have addressed all of my comments.

Reviewer #2 (Remarks to the Author):

As it is a simple physical calculation the authors could well indicate the width of their light sheet at the edges of their field of view if they know the thickness at the focus. I regard this as important as often only the light sheet thickness is given at the thinnest point and then biologists think this is the dimension of the light sheet everywhere.

if done so the paper can be published

Reviewer #3 (Remarks to the Author):

No further concerns

REVIEWERS' COMMENTS

Reviewer #1 (Remarks to the Author):

The authors have addressed all of my comments.

Reviewer #2 (Remarks to the Author):

As it is a simple physical calculation the authors could well indicate the width of their light sheet at the edges of their field of view if they know the thickness at the focus. I regard this as important as often only the light sheet thickness is given at the thinnest point and then biologists think this is the dimension of the light sheet everywhere.
if done so the paper can be published

We thank the reviewer for their suggestion. We have estimated the width of the light-sheet at the center and edges of the tile using the equations below^{1,2}. We assumed that the light-sheet followed a Gaussian beam profile with a low beam divergence angle.

Calculation of the beam waist at the center of the tile:

$$w_0 = \frac{2w}{k} \quad (1)$$

$$w = \frac{2w_0}{k} \quad (2)$$

Variables: w_0 is the beam waist (one-half of the light-sheet width), k is the angular wavenumber in the medium (TDE), n is the refractive index of the medium (TDE), NA is the numerical aperture, and λ is the wavelength in the medium (TDE)

Calculation of the beam waist at the edge of the tile:

$$w_z = w_0 \sqrt{1 + \left(\frac{Z}{Z_R}\right)^2} \quad (3)$$

$$Z_R = \frac{\lambda}{\pi w_0^2} \quad (4)$$

Variables: w_z is the beam waist (one-half of the light-sheet width) at a distance Z away from the center of the tile, Z_R is the Rayleigh length, w_0 is the beam waist at the center of the tile, Z is the distance between the center and edge of the tile, and λ is the wavelength in the medium (TDE)

We calculated the light-sheet width based upon the following parameters of our imaging setup:

- Numerical aperture (NA): 0.154
- Vacuum wavelength of the light-sheet (λ_{vac}): 561 nm – 785 nm
- Refractive index of TDE (n): 1.5215
- Distance between center and edge of the tile (Z): 1404 μm

Width of the light-sheet at the center of each tile: 2.3 μm – 3.2 μm (calculated using Eqs. (1) and (2))

Width of the light-sheet at the horizontal edges of each tile: 284.2 μm (calculated using Eqs. (3) and (4), value is consistent across the range of wavelengths)

We have modified the “*Light-sheet imaging*” section of the Methods to include the estimated light-sheet width:

The following hardware and settings were used for all scans: 2.5X zoom with a 2X dipping cap (5X magnification, 1.3 μm x-y pixel size), 5.5 Megapixel sCMOS camera, 20 ms exposure time,

0.154 numerical aperture, and 2.5 μm z step size. Based upon the assumption that the light-sheet followed a Gaussian beam profile, the estimated light-sheet width was 2.3-3.2 μm at the center of each tile and 284.2 μm at the horizontal edges of each tile for all scans.

References:

1. Novotny, L. & Hecht, B. Propagation and focusing of optical fields. in *Principles of Nano-Optics* 45–88 (Cambridge University Press, 2006). doi:10.1017/CBO9780511813535.004
2. Gaussian beams, explained by RP Photonics Encyclopedia; laser beam, fundamental transverse modes. Available at: https://www.rp-photonics.com/gaussian_beams.html. (Accessed: 10th August 2021)

Reviewer #3 (Remarks to the Author):

No further concerns